# PILLET-GAN: Pixel-Level Lesion Traversal Generative Adversarial Network for Pneumonia Localization

**HyunWoo Kim**[1]                                                                 HYUNWOO.KIM@VUNO.CO
**HanBin Ko**[1]                                                                      HANBIN.KO@VUNO.CO
**JungJun Kim**[2]                                                                JAYDEN@DEEPBRAINAI.IO
[1] *VUNO Inc, Seoul, Korea*
[2] *DeepBrain AI, Seoul, Korea*

## Abstract

The study of pneumonia localization focus on the problem of accurate lesion localization in the thoracic X-ray image. It is crucial to provide precisely localized regions to users. It can lay out the basis of the model decision by comparing the X-ray image between the 'Healthy' and 'Disease' classes. In particular, for the medical image analysis, it is essential not only to make a correct prediction for the disease but also to provide evidence to support accurate predictions. Many generative adversarial networks (GAN) based approaches are employed to show the pixel-level changes via domain translation technique to address this issue. Although previous research tried to improve localization performance by understanding the domain's attributes for better image translation, it remains challenging to capture the specific category's pixel-level changes. For this reason, we focus on the stage of understanding the category attributes. We propose a Pixel-Level Lesion Traversal Generative Adversarial Network (PILLET-GAN) that mines spatial features for the category via spatial attention technique and fuses them into an original feature map extracted from the generator for better domain translation. Our experimental results show that PILLET-GAN achieves superior performance compared to the state-of-the-art models on qualitative and quantitative results on the RSNA-pneumonia dataset.

**Keywords:** Medical image analysis, Generative adversarial network, Pneumonia localization, Computer-aided system

## 1. Introduction

Along with the development of deep learning techniques, the demand for explainable artificial intelligence (XAI) study has risen above the surface in the area of Computer Vision (CV) and Natural Language Processing (NLP). In particular, the XAI system is essential in the medical domain. The doctors, user of medical AI, do not rely entirely on the predictions of the deep learning model rather using it as an auxiliary tool. In the result, it is essential to provide reasonable basis along with the prediction results.

Meanwhile, the initial works of XAI propose a method that visualizes the model's decision by using class activation map or posterior gradient information for the task of classification (Zhou et al., 2016; Selvaraju et al., 2017; Chattopadhay et al., 2018). However, the main limitation of this method is that it visualizes the model's decision evidence rather extensively. Classification-based localization approaches are vulnerable of providing high-quality visual evidence for multi-regional areas. This is because the classifier itself filters

out relatively fewer discriminative features within the image by learning only the most discriminative area for accurate prediction. To address this issue, the GAN-based localization approach(Baumgartner et al., 2018; Siddiquee et al., 2019; Schlegl et al., 2017, 2019; Gong et al., 2019) has attracted much attention in CV communities. It is designed to initially learn the class distribution and then generate synthetic images with the target domain's attributes. For localization, GAN presents visual attributes for synthesizing into target domains through differences between synthesized and original images. However, since GAN focuses on generating realistic images rather than synthesizing the attributes, heavy amount of noise is inevitable when using the difference between two images. In GAN-based localization, this noise is a critical problem because it leads to false predictions. To address this issue, we propose a Pixel-Level Lesion Traversal Generative Adversarial Network (PILLET-GAN) to localize the pixel-level lesion's area without ground truth label. We present a method of synthesizing important attributes into the target domain from the generator's feature maps through the spatial attention block and the mixing block. Our experimental results show significant improvements in localization tasks compared to the state-of-the-art models on the RSNA-pneumonia dataset. In summary, our contributions of this work are as follows:

1) We present a pixel-level localization method to remove the feature map's noise and accurately synthesize disease areas by using two sub-networks including the spatial attention block and the mixing block.

2) We show for the first time pixel-level visualization of pneumonia disease in the chest X-ray domain using a generative model.

3) We demonstrate the effectiveness of our method on the RSNA-pneumonia dataset with various state-of-the-art models.

## 2. Related Work

### 2.1. Localization approaches in Medical domain

Localization task for medical image analysis is a task of pointing out the lesions that relates to the diseases' diagnosis. It is required for the localization results to be specific and clear in that they present explicit evidence, not extensive. Initially, classification-based methods (Zhou et al., 2016; Selvaraju et al., 2017) were used to be compatible with the previous classification models. (Zhou et al., 2016) propose a class activation map (CAM) to provide a classifier's decision evidence that visualizes the feature maps where the activated maps are calculated by multiplying the last feature map from the CNN model with the weights of the fully-connected layer. (Selvaraju et al., 2017) propose Grad-CAM that uses gradient information to reflect the model's decision in detail. Unlike CAM, Grad-CAM can have the more flexible structure of the classification model in that they only need gradient vectors for prediction. However, these approaches do not consider an image's spatial information in detail, and it has limited use in medical image analysis domain. To address this problem, various GAN-based localization methods (Baumgartner et al., 2018; Seah et al., 2019; Siddiquee et al., 2019; Taghanaki et al., 2019; Zhang et al., 2019) have been proposed for medical image analysis. In particular, (Siddiquee et al., 2019) proposed Fixed-Point GAN

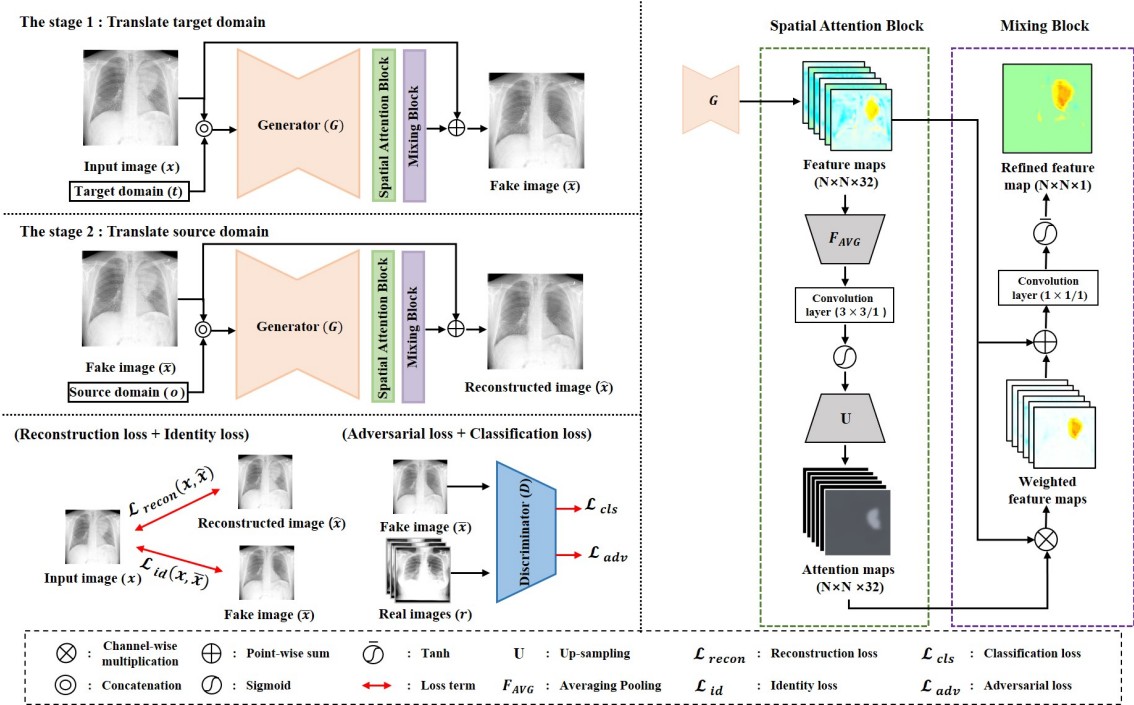

Figure 1: PILLET-GAN consists of two sub-networks: spatial attention block and mixing block. The spatial attention block extracts meaningful features to reduce noises. The mixing block then preserves original feature information by aggregating the attention map and output feature maps from the generator.

for detection and localization of the diseased region. (Siddiquee et al., 2019) introduced a conditional identity loss to preserve the identity of an original image for better image translation. However, Fixed-Point GAN mainly considers maintaining the input image label, which undesirable changes still occur during domain translation. Previous approaches have focused on inter-domain translation. For precise pixel-level localization, it is necessary to consider inter-domain translation along with visual attributes of the category which are relatively more important. We propose a method that efficiently collects category's visual attributes and utilizes the original features to obtain refined areas.

## 3. Model

In this section, we present Pixel-Level Lesion Traversal Generative Adversarial Network (PILLET-GAN) for pixel-level pneumonia localization. We train GAN to translate the input X-ray image into the target domain and then localize the pneumonia area through obtaining the difference between the input X-ray image and the 'Healthy' domain X-ray image synthesized from the generator. Our network consists of two sub-networks to enhance the network's attention power, as shown in Figure 1. First, spatial attention block is designed to reduce unnecessary pixel shifts from the synthesized image. Second, the mixing

block are used to output attention map that is extracted from spatial attention block to compensate the original information of the class with the residual connection. Finally, by using the two mentioned sub-blocks, we obtain a fine-grained localization map that indicates pneumonia regions.

### 3.1. Spatial Attention Block

The spatial attention block adjusts the transformation area to provide guidance on the generator. Given feature maps $G'$ from the generator, the spatial attention block is shown in Eq. (1) and (2).

$$SAB\left(G'\right) = U\left(\sigma\left(ConvNet\left(F_{avg}^{4\times4/4}\left(G'\right)\right)\right)\right) \tag{1}$$

$$ConvNet\left(\cdot\right) = f_{conv}^{3\times3/1}\left(\delta\left(IN\left(f_{conv}^{3\times3/1}\left(\cdot\right)\right)\right)\right), \tag{2}$$

where $F_{avg}^{4\times4/4}$ represents average pooling with kernel size 4 and stride 4, and $f_{conv}^{3\times3/1}$ represents a convolution network with kernel size 3 and stride 1. We use instance normalization $IN(\cdot)$ to eliminate style variations (Ulyanov et al., 2016). $\delta$ and $\sigma$ denote ReLU and Sigmoid, respectively. $U$ is the up-sampling operation. To sum up, we generate a spatial attention map by considering inter-spatial relationship of the feature maps. In particular, we use average pooling to integrate the neighboring spatial information followed by each down-sampled value becoming the representative value of the grid. Then, we use a convolution network to train the inter-dependency within representative values. We generate a probability map by applying the sigmoid function. The final feature map of the spatial attention block is up-sampled to match the size of $G'$.

### 3.2. Mixing Block

After applying the spatial attention block to $G'$, a residual connection is used to maintain the well-learned attributes of the original features. We then reduce the dimension of the feature maps using a 1×1 convolution network in order to consider channel-to-channel correlation information. By applying a non-linear activation function, we obtain a lesion map $L_{Mixed}$, and a fake image $x_{fake}$ which is generated by combining the lesion map and the original image. This process is formally expressed as follows:

$$L_{Mixed} = tanh\left(f_{conv}^{1\times1/1}\left(\left(SAB\left(G'\right)\oplus1\right)\otimes G'\right)\right) \tag{3}$$

$$x_{fake} = x_{origin}\oplus L_{Mixed} \tag{4}$$

We found that the attention module enhances localization power in the lesion map while maintaining the quality of the synthesized image by reducing unnecessary changes.

### 3.3. Training Objectives

#### 3.3.1. ADVERSARIAL LOSS

GAN proposed by (Goodfellow et al., 2014) shows excellent performance in the task of image-to-image translation. However, limitations exists, such as mode collapse that pro-

duces only one image to deceive discriminators and saturated gradients that backwards meaningless gradients. Thus, we substitute GAN's loss functions into WGAN (Arjovsky et al., 2017) and WGAN-GP (Gulrajani et al., 2017) as expressed in Eq. (5) and (6).

$$L_{adv}^{G} = -\mathbb{E}_{x,t}\left[D_{r/f}\left(G\left(x,t\right)\right)\right] \tag{5}$$

$$L_{adv}^{D} = \mathbb{E}_{x}\left[D_{r/f}\left(x\right)\right] - \mathbb{E}_{x,t}\left[D_{r/f}\left(G\left(x,t\right)\right)\right] - \lambda_{gp}\mathbb{E}_{\tilde{x}}\left[\left(||\nabla_{\tilde{x}}D_{r/f}\left(\tilde{x}\right)||_{2} - 1\right)^{2}\right], \tag{6}$$

where $x$ is the input X-ray image and $t$ is the target domain vector. $\lambda_{gp}$ a hyper-parameter that represents the gradient penalty. $\tilde{x}$ is a sampled point from the linearly interpolated distribution between the sampled distribution of the real data and the generated distribution of the synthesized data. The gradient penalty term is added for the discriminator to approximate 1-lipschitz constraint (Gulrajani et al., 2017). $D$ represents a discriminator, and $G$ represents a generator containing two sub-networks.

### 3.3.2. RECONSTRUCTION LOSS

The reconstruction loss is applied to help the generator transform only domain-specific conditions while maintaining the shape of the input image as follows:

$$\mathcal{L}_{recon} = \mathbb{E}_{x,t}\left[|| G\left(G\left(x,t\right),o\right) - x ||_{1}\right], \tag{7}$$

where $o$ represents the original domain vector of the input image $x$, and $t$ represents the target domain vector which we want to synthesize.

### 3.3.3. IDENTITY LOSS

We use the identity loss to change only the most necessary parts representing a domain. Eq. (8) shows our identity loss function that consists of two terms:

$$\mathcal{L}_{id} = \mathbb{E}_{x,t}\left[|| G\left(x,t\right) - x ||_{1}\right] + \mathbb{E}_{x,o}\left[|| G\left(x,o\right) - x ||_{1}\right] \tag{8}$$

The first term is the target identity loss, represented as the L1 difference between the input image $x$ and the generated image $G(x,t)$. The second term is the original identity loss, which is used by the generator to preserve the identity of the original domain.

### 3.3.4. CLASSIFICATION LOSS

The classificatio loss to control the domain of the generated fake image is as follows:

$$\mathcal{L}_{cls}^{real} = \mathbb{E}_{x,o}\left[-\log D_{cls}\left(o|x\right)\right] \tag{9}$$

$$\mathcal{L}_{cls}^{fake} = \mathbb{E}_{x,t}\left[-\log D_{cls}\left(t|G\left(x,t\right)\right)\right] \tag{10}$$

Using Eq. (9), the discriminator learns the distribution of real data. Eq. (10) encourages the generator to produce a fake image with the target domain vector to approximate the real data distribution using the discriminator.

### 3.3.5. Oᴠᴇʀᴀʟʟ Lᴏss

The total loss function of the discriminator and the generator is defined as follows:

$$\mathcal{L}_D = -\mathcal{L}_{adv}^{D} \; + \; \lambda_{cls}\mathcal{L}_{cls}^{real} \tag{11}$$

$$\mathcal{L}_G = \mathcal{L}_{adv}^{G} \; + \; \lambda_{cls}\mathcal{L}_{cls}^{fake} + \lambda_{recon}\mathcal{L}_{recon} \; + \; \lambda_{id}\mathcal{L}_{id}, \tag{12}$$

where $\lambda_{cls}$, $\lambda_{recon}$, $\lambda_{cid}$ are hyperparameters that control each loss term.

## 4. Experiments

### 4.1. Datasets

Our model is trained and evaluated on the RSNA-pneumonia dataset. The RSNA-pneumonia dataset is composed of 14,862 frontal view X-ray images with or without the thoracic disease. We divide the training set and validation set by 9:1 ratios. Both training set and validation set contain each 13,386 images (Pneumonia: 5,443, Healthy: 7,943) and 1,487 images (Pneumonia: 579, Healthy: 908).

### 4.2. Implementation and Training Details

#### 4.2.1. Iᴍᴘʟᴇᴍᴇɴᴛᴀᴛɪᴏɴ

We optimize PILLET-GAN using the Adam (Kingma and Ba, 2014) optimizer with a batch size of 8. We set the learning rate of the generator and discriminator as 1e-4. We normalize the input X-ray image from [0, 255] to [-1,1]. We set $\lambda_{cls}$=1, $\lambda_{recon}$=10, $\lambda_{id}$=10 and $\lambda_{gp}$=10 for the hyperparameter of each classification loss, reconstruction loss, identification loss and gradient penalty loss. For stable learning, we balance the learning of generators and discriminators by the ratio of 1:5 to have the generator quickly learn from the meaningful gradient of the discriminator. We train PILLET-GAN for about 100K iterations and took about 10 hours on a single Titan Xp GPU.

#### 4.2.2. Tʀᴀɪɴɪɴɢ Dᴇᴛᴀɪʟs

Our model is built upon the baseline StarGAN model (Choi et al., 2018). We train our generator with adversarial loss, classification loss, identity loss, reconstruction loss and discriminator with adversarial loss and classification loss on the RSNA-pneumonia dataset. We train the generator by randomly setting the target domain so that the generator can learn the distribution of disease and healthy domains. In the test phase, we perform localization using the lesion map, and the target domain is fixed to the 'health' class.

### 4.3. Comparison with State-Of-The-Arts Models

#### 4.3.1. Qᴜᴀʟɪᴛᴀᴛɪᴠᴇ Rᴇsᴜʟᴛs

We compare PILLET-GAN to a classifier-based localization; CheXNet-CAM (Rajpurkar et al., 2017) and generative model-based localization methods; VA-GAN (Baumgartner et al., 2018), StarGAN (Choi et al., 2018) and Fixed-Point GAN (Siddiquee et al., 2019).

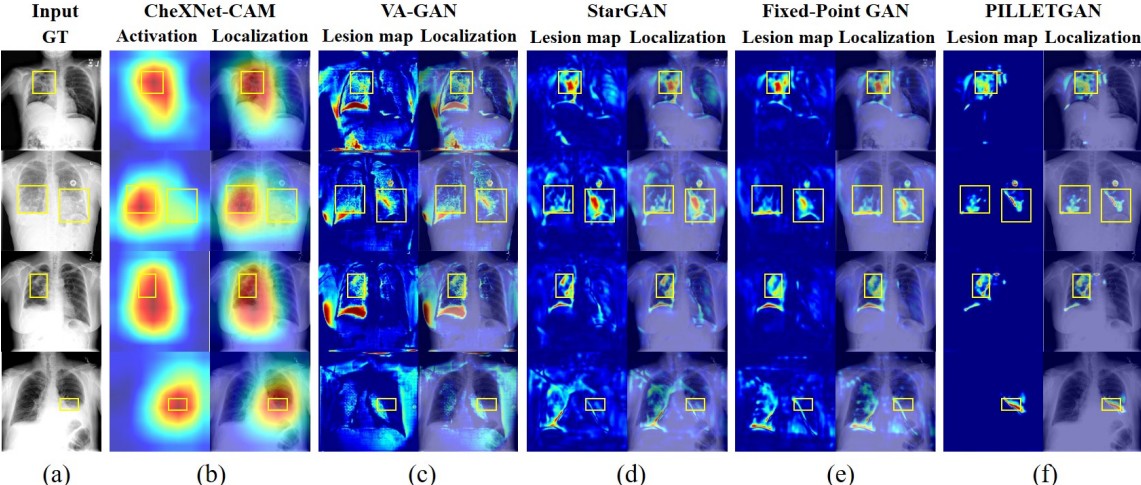

| Input GT | CheXNet-CAM | | VA-GAN | | StarGAN | | Fixed-Point GAN | | PILLETGAN | |
|---|---|---|---|---|---|---|---|---|---|---|
| | Activation | Localization | Lesion map | Localization | Lesion map | Localization | Lesion map | Localization | Lesion map | Localization |

(a)   (b)     (c)     (d)     (e)     (f)

Figure 2: Qualitative evaluation of localization performance in various methods. (a) Chest X-ray images with ground truth boxes. (b) Classifier-based method (CheXNet). (c)-(f) Generative model-based methods including PILLET-GAN.

We show the localization performance using pneumonia images along with the ground-truth boxes in Figure. 2. For the evaluation of localization, we calculated the normalized difference map between a synthesized image from PILLET-GAN and the input X-ray image. We observe that CheXNet-CAM($2^{nd}$ column) does visualize a wide range of areas rather than visualizing multi-regions. Other models show poor performance in visualizing lesions due to unnecessary changes. On the other hand, PILLET-GAN shows a more precise localization of the disease area guided by the spatial attention block. More qualitative results can be found in the appendix section.

### 4.3.2. QUANTITATIVE RESULTS

For quantitative evaluation, we use an accuracy metric to evaluate the quality of synthesized images using a pretrained CheXNet-CAM (Rajpurkar et al., 2017) network. We also measure the pixel-change rate to measure the quantity of changed pixels for image translation. The pixel-change rate is defined as

$$\text{Pixel change rate} \quad = \quad \sum_{ij} \frac{d_{ij}(x)}{WH}, \quad \text{where} \quad d_{ij}(x) \quad = \quad \begin{cases} 0, \text{ if } x_{ij} = G_{ij}(x,t) \\ 1, \text{ otherwise.} \end{cases} \quad (13)$$

Tables 1, 2 show the accuracy and pixel change rates respectively. In the tables, '$X \to Y$' denotes that images belong to domain '$X$' are synthesized to images corresponding to domain '$Y$'. P, H, and P+H represent domains labeled 'Pneumonia', 'Healthy', and the entire test data set, respectively. PILLET-GAN not only translates fewer pixels compared to other generative models for the synthesis but also maintains similar accuracy compared to others. Lastly, we used the localization metric, AUC, that evaluates how well the predicted

Table 1: Accuracy using the pretrained CheXNet network after synthesizing an input image to the 'Healthy' domain. evaluation.

|  | VA-GAN | StarGAN | Fixed-Point GAN | PILLET-GAN (w/o SAB) | PILLET-GAN |
|---|---|---|---|---|---|
| P → H | 75.34% | 85.31% | 88.42% | 88.42% | 87.56% |
| H → H | 98.56% | 99.55% | 99.44% | 99.66% | 98.12% |
| P+H → H | 86.82% | 94.01% | 95.15% | 95.29% | 94.01% |

Table 2: Pixel change rate when synthesizing an input image to the 'Healthy' domain. PILLET-GAN modifies smaller areas than others in image-to-image translation.

|  | VA-GAN | StarGAN | Fixed-Point GAN | PILLET-GAN (w/o SAB) | PILLET-GAN |
|---|---|---|---|---|---|
| P → H | 0.7639 | 0.6212 | 0.5226 | 0.2391 | 0.2612 |
| H → H | 0.6995 | 0.3958 | 0.2425 | 0.1474 | 0.1061 |
| P+H → H | 0.7247 | 0.4836 | 0.3515 | 0.1831 | 0.1664 |

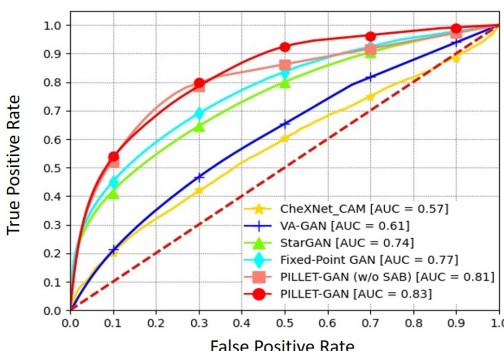

Figure 3: ROC curve on the RSNA-pneumonia dataset. Our model has the highest AUC value (0.83) on the image-level localization. Best viewed in color.

area fits in to the ground truth (Taghanaki et al., 2019). This can be done by calculating the ratio of overlapping pixels with the ground truth boxes. As shown in Figure 3, our method outperforms others in terms of the localization performance with 0.83 AUC.

## 5. Conclusion

In this work, we present an approach for pixel-level localization of pneumonia disease. Our method removes the feature map's noise and accurately sunthesize diseased areas by using two sub-networks including the spatial attention block and the mixing block. Our quantitative results show that PILLET-GAN effectively improves the performance of localization on the RSNA-pneumonia dataset. Besides, our qualitative results show that our localized results are superior compared to both classification-based models and GAN-based models.

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

## Appendix A. Comparison results of changed pixel area

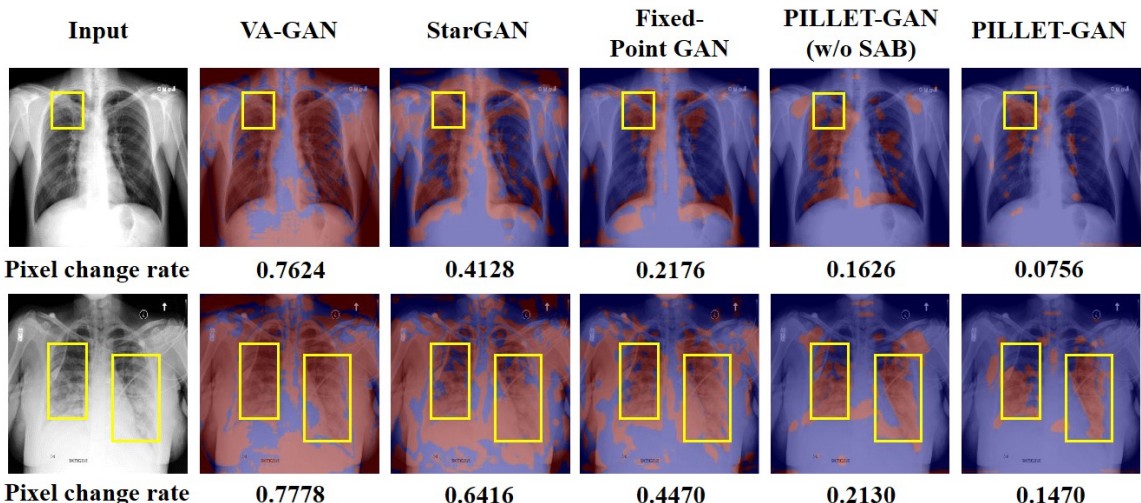

Figure 4: Comparison results of changed pixel area when synthesized from 'Pneumonia' to 'Healthy'. Compared to other methods, PILLET-GAN shows that focusing more on the lesion area than other networks because it tends to synthesize the essential parts during the generative process.

# Appendix B. Success and failure cases with our PILLET-GAN

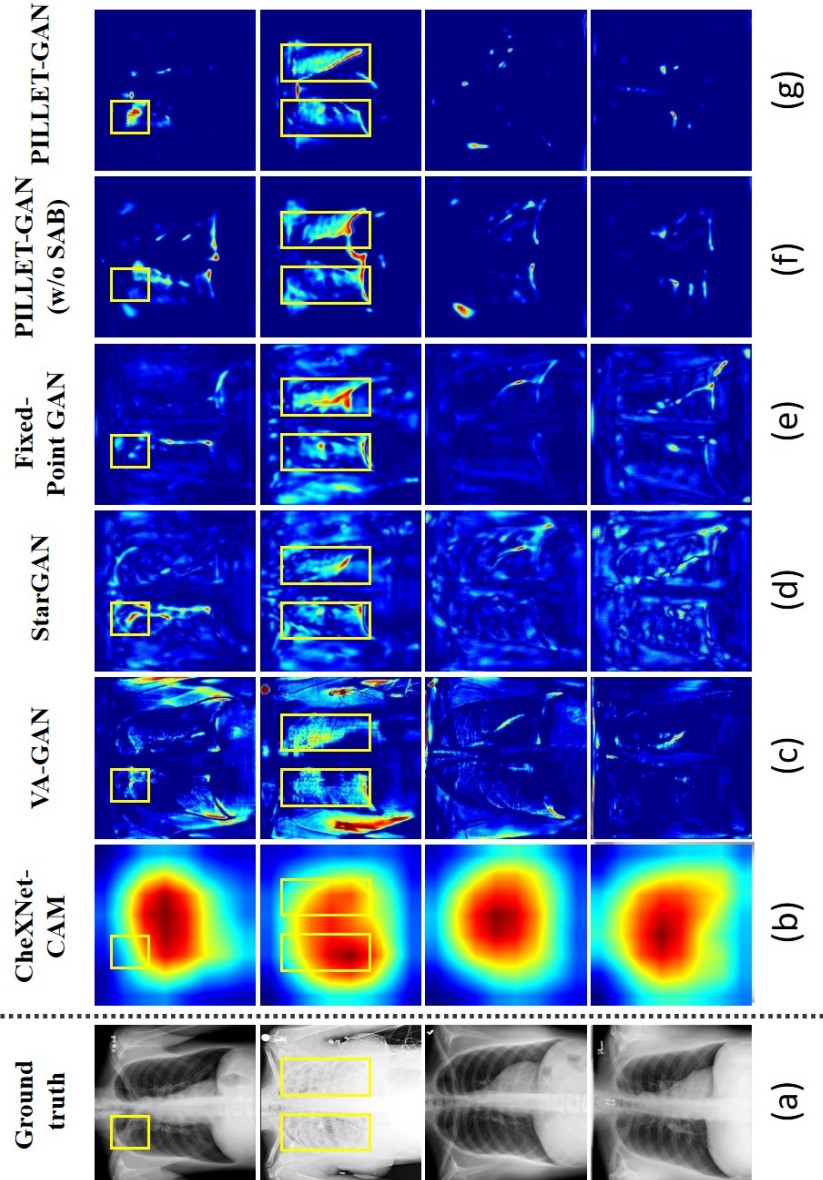

Figure 5: Successful cases with our PILLET-GAN. (a) Input x-ray images. (b) The activation map using a classifier-based method(CheXNet). (c)-(g) The lesion map using generative model-based methods. 1st and 2nd rows: Synthesized from "Pneumonia" to "Healthy". 3nd and 4nd rows: Synthesized from "Healthy" to "Healthy". Our meth-od can localize legions correctly even when an image looks similar to normal (e.g., 1st row versus 3nd or 4nd rows).

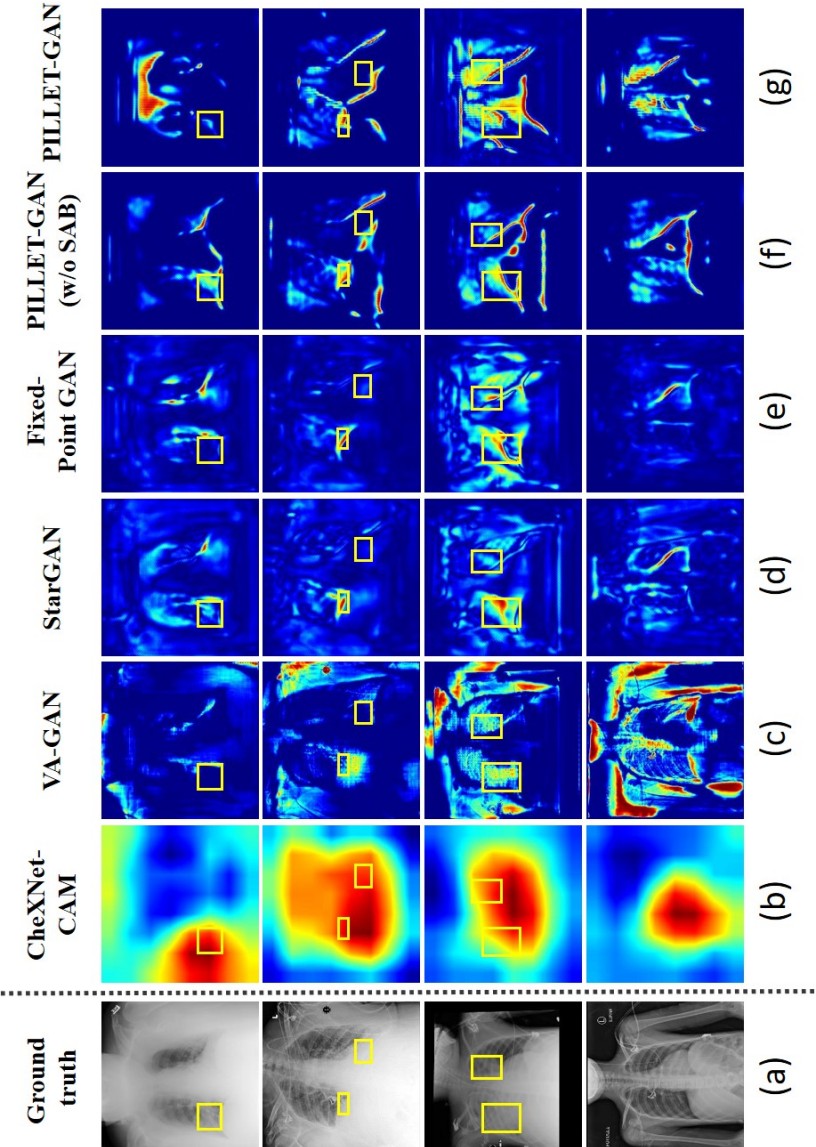

Figure 6: Failure cases of all localization methods. (a) Input x-ray images. (b) The activation map using a classifier-based method(CheXNet). (c)-(g) The lesion map using genera-tive model-based methods. 1st–3nd rows: Synthesized from "Pneumonia" to "Healthy". 4nd row: Synthesized from "Healthy" to "Health". Some cases such as thoracic images containing gas or fluid, pediatric chest disease, whole body chest images fail to produce correct lesion localization for all methods due to the limited data distribution of the RSNA-pneumonia dataset.

