# OpenReview forum: "PILLET-GAN: Pixel-Level Lesion Traversal Generative Adversarial Network for Pneumonia Localization"
_MIDL.io/2022/Conference — MIDL 2022_

### Official Review · Reviewer_G1d1 · 2022-01-19

**Confidence:** 4
**Preliminary Rating:** 4
**Recommendation:** Poster

**Summary:**

The authors present a gan-based methodology for localizing pneumonia on X-Ray scans. The method is an extension of Siddiquee et al., 2019 in that it adds two extra blocks on top of the generator, namely the attention and mixing blocks. The blocks are claimed to help remove noise (often coming during inter-domain translation).

**Strengths:**

The method seems to perform well both qualitatively, Fig. 2, and quantitatively, Fig. 3., beating the SOTA for the task of localization.
The manuscript is well written, with a thorough intro to the topic and related literature, and a presentation of the results.

**Weaknesses:**

The method is essentially a close replica of the Siddiquee et al., 2019 equipped with known ingredients of the deep learning alchemy's, i.e. attention, and the mixing block relying on residual connections. So methodological development is quite incremental. But it provides an increase in performance compared to SOTA, so for MIDL, it is probably good enough.


**Deanonymize Review:**

no

**Final Rating After The Rebuttal:**

4: Weak Accept

**Justification Of The Final Rating:**

Thanks for the rebuttal. I keep my rating as described originally due to rather limited methodological contribution.
I keep my rating as described originally due to rather limited methodological contribution.

**Paper Type:**

methodological development

**Questions To Address In The Rebuttal:**

I gave "weak accept" as I regard the methodological contribution as moderate, even though the method improves the performance for the localization task. Apart from this, I am content with the current version of the manuscript.

**Special Issue:**

no

---

### Official Review · Reviewer_gvwD · 2022-01-24

**Confidence:** 4
**Preliminary Rating:** 4
**Recommendation:** Poster

**Summary:**

This paper presents a method for pneumonia localization in the thoracic X-ray images by training networks with only image-level classification label. The method is based on generative adversarial network (GAN) to translate between the pneumonia and healthy images, and utilizes the differences between a translated image and original image to indicate the lesion areas. A spatial attention block and mixing block are incorporated into GAN to guide the translation to focus on the disease areas. Evaluation is conducted on the RSNA-pneumonia dataset with improved performance obtained comparing to other localization methods.

**Strengths:**

- The paper is clearly written and easy to read;
- The authors present a well-motivated method for pneumonia localization.
- The experimental evaluation validates the effectiveness of the method with improved performance demonstrated.

**Weaknesses:**

- The main technical novelty is the attention mechanism achieved by the proposed spatial attention block and the mixing block, but the contributions of the two modules are not clearly demonstrated. The authors should have an ablation study on the key components.

- For the training objectives, various loss items are utilized simultaneously in the GAN framework. The benefits and necessity of each loss item are not clearly introduced. For example, what are the purposes of including the identity loss and the classification loss? What would be the results if removing either loss item?


**Deanonymize Review:**

no

**Final Rating After The Rebuttal:**

4: Weak Accept

**Justification Of The Final Rating:**

The reviewer would like to keep the original score since as a methodological development paper, the methodological contribution is not very significant. The justification for including the identify loss and classification loss is not clearly provided.

**Paper Type:**

methodological development

**Questions To Address In The Rebuttal:**

- The authors should justify the importance of the proposed spatial attention block and mixing block;
- The authors should introduce and evaluate the contribution of each loss item more clearly, especially for the identity loss and the classification loss.

**Special Issue:**

no

---

### Meta-Review · Area_Chair_7PXH · 2022-02-19

**Recommendation:** Accept (Poster)
**Confidence:** 5

**Metareview:**

The presented work received one borderline and two weak accepts pre-rebuttal, which remained unchanged after the rebuttal phase. In particular, despite the authors' responses, the reviewers felt that the methodological contribution is rather limited, which they consider as an important weakness of the proposed work. While I agree that the technical novelty is marginal, there might exist other sources of 'novelty', for example in terms of application, which seems to be the case in this work. Following the reviewers scores, and considering that the authors have positively addressed most of the raised comments, I recommend the acceptance of this paper as a poster.

---

### Decision · Program_Chairs · 2022-02-28

Accept